# Arginine Reduces Glycation in γ_2_ Subunit of AMPK and Pathologies in Alzheimer’s Disease Model Mice

**DOI:** 10.3390/cells11213520

**Published:** 2022-11-07

**Authors:** Rui Zhu, Ying Lei, Fangxiao Shi, Qing Tian, Xinwen Zhou

**Affiliations:** Key Laboratory of Neurological Disease of Education Ministry, Department of Pathophysiology, Tongji Medical College, Huazhong University of Science and Technology, Wuhan 430030, China

**Keywords:** Alzheimer’s disease, glycation, advanced glycation end−products, AMPK, L−arginine

## Abstract

The metabolism disorders are a common convergence of Alzheimer’s disease (AD) and type 2 diabetes mellitus (T2DM). The characteristics of AD are senile plaques and neurofibrillary tangles (NFTs) composed by deposits of amyloid−β (Aβ) and phosphorylated tau, respectively. Advanced glycation end−products (AGEs) are a stable modification of proteins by non−enzymatic reactions, which could result in the protein dysfunction. AGEs are associated with some disease developments, such as diabetes mellitus and AD, but the effects of the glycated γ_2_ subunit of AMPK on its activity and the roles in AD onset are unknown. Methods: We studied the effect of glycated γ_2_ subunit of AMPK on its activity in N2a cells. In 3 × Tg mice, we administrated L−arginine once every two days for 45 days and evaluated the glycation level of γ_2_ subunit and function of AMPK and alternation of pathologies. Results: The glycation level of γ_2_ subunit was significantly elevated in 3 × Tg mice as compared with control mice, meanwhile, the level of pT172−AMPK was obviously lower in 3 × Tg mice than that in control mice. Moreover, we found that arginine protects the γ_2_ subunit of AMPK from glycation, preserves AMPK function, and improves pathologies and cognitive deficits in 3 × Tg mice. Conclusions: Arginine treatment decreases glycated γ_2_ subunit of AMPK and increases p−AMPK levels in 3 × Tg mice, suggesting that reduced glycation of the γ_2_ subunit could ameliorate AMPK function and become a new target for AD therapy in the future.

## 1. Introduction

Alzheimer’s disease (AD), which is a progressive neurodegeneration disease, is the most common cause of dementia; by 2030, there will be more than 49 million people suffering from AD [1]. Patients’ symptoms worsen over the course of AD and present as dysfunctions in memory, orientation, judgment, problem−solving, involvement in community affairs, and so on [2]. There are two typical neuropathological characteristics reported in AD, amyloid−β (Aβ) peptides in extracellular senile plaques and intracellular neurofibrillary tangles (NFTs) composed of hyper−phosphorylation of microtubule−association protein Tau [2,3]. Type 2 diabetes mellitus (T2DM) is considered as an independent risk factor for AD and shares lots of similar pathologies with AD [4]. Patients with AD exhibit abnormalities in the brain’s glucose metabolism, insulin resistance, etc. [5,6]. The glucose uptake in the brain reduces before Abeta or NFTs. Disruptions in insulin and insulin−like growth factor 1(IGF−1) signaling impairs neuronal energy metabolism, survival, and so on. Patients with T2DM may be more susceptible to learning and memory malfunctions than individuals without T2DM. Insulin receptor is widely distributed throughout the brain: therefore, periphery hyperinsulinemia affects brain function, such as synaptic plasticity and neurotransmitters. Chronic hyperglycemia is closely associated with cerebral vascular damages, ROS, inflammation, etc. Moreover, the imbalance of energy metabolism results in glucose metabolic by−products, especially advanced glycation end−products (AGEs) [7,8,9]. Furthermore, AGEs contribute to insulin resistance, formations of Aβ depositions, and NFTs [10,11].

The 5′−adenosine monophosphate (AMP) −activated protein kinase (AMPK) is a conserved serine/threonine kinase formed by hetero-trimeric proteins and catalytic α− and regulatory β− and γ−subunits. AMPK is an energy sensor maintaining energy homeostasis by suppressing anabolic metabolism while promoting catabolic metabolism [12]. The phosphorylation of α− subunit of AMPK at Thr172 site dramatically increases the kinase activity. The β− subunit contains a carbohydrate−binding module (CBM), which plays a crucial role in the temporospatial regulation of AMPK signaling [13]. The regulation of AMPK activity is described at three different levels: ① activation loop phosphorylation by upstream kinases; ② protection against activation loop de−phosphorylation by protein phosphatases; ③ allosteric kinase activation [14,15]. Accumulating evidence has shown that the γ−subunit of AMPK sensing the ratio of adenosine triphosphate (ATP) to AMP/ADP allosterically regulates AMPK, but the mechanisms are still unclear [15]. The γ−subunit of AMPK contains a conserved adenine nucleotide−binding domain that contains four cystathione β−synthetase (CBS) AMP/ADP/ATP binding sites and CBS3 effects on the regulation of AMPK activity by AMP/ATP [16]. AGEs result from a series of non−enzymatic reactions, called Maillard reactions, which react between N−terminal amino acid residues and/or ε−amino groups of proteins and carbonyl groups of reducing sugars or their derivatives [17]. Methylglyoxal (MG) as a by−product of the glycolytic pathway derives from the degradation of triose phosphates or non−enzymatically by sugar fragmentation reactions and is the most potent precursor of AGEs [18]. MG has a binding preference toward arginine residues [19]. More and more evidence has shown that glycated arginine or lysine will alter or even disrupt the structure of proteins and affect their function [20,21,22], which is the same as a mutation in amino acid residues. Thus, we supposed that arginine residues glycated by MG or its analogs at adenine nucleotide−binding in the γ−subunit of AMPK will disrupt the regulation of AMPK activity and induce AD−like pathologies.

Many researchers, including us, have demonstrated that activating AMPK exerts broad neuroprotective effects [23,24,25]. The brain, which is a heightened metabolic organ, is extremely vulnerable to glucose abnormalities, especially in the hippocampus [26]. Much evidence has shown decreased glucose uptake as well as the availability of ATP even in the early stage of AD while activating AMPK compensates for the loss of ATP [27]. What is more, AMPK decreases Aβ level by decreasing its production and enhancing the clearance [28,29,30], like the hyper-phosphorylation of Tau [31]. Furthermore, activation of AMPK alleviates chronic inflammation in the brain [32]. On the other hand, it is reported that AMPK activity is impaired in the hippocampal of APP/PS1 mice [33], but the mechanism of disruption of AMPK is unknown in AD.

In the current study, we found that the level of pT172−AMPK (activation) was obviously decreased in 3 × Tg mice; meanwhile, the γ_2_ subunit of AMPK as well as of brain proteins was highly glycated. On the other hand, we discovered that arginine (Arg) protects γ_2_ subunit of AMPK from glycation. Intraperitoneal injection of arginine inhibited glycation of γ_2_ subunit of AMPK in the hippocampus, improved activity of AMPK, and rescued cognitive dysfunction in 3 × Tg mice.

## 2. Materials and Methods

### 2.1. Antibody and Drugs

Rabbit polyclonal antibodies (pAb) AMPKα, pThr172−AMPK, ACC, pSer79−ACC, AMPK γ_2_ (1:100 for immunoprecipitation), APP (1:500 for western blot), and a mouse monoclonal antibody (mAb) against Tau phosphorylated at Ser396 (1:1000 for western blot) were purchased from Cell Signal Technology. Rabbit pAb against AGEs (1:100 for immunoprecipitation, 1:200 for immunohistochemistry, 1:500 for western blot and dot blot) and mouse mAb Tau 5 against total tau (1:1000 for western blot) were purchased from Abcam. Rabbit pAb against Tau phosphorylation at Ser404 (1:1000 for western blot) was purchased from SAB. A mAb against β−actin (1:1000 for western blot), MG (M0252), and L−arginine (11009–100G−F) were purchased from Sigma Aldrich. Dulbecco’s modified Eagle’s medium (DMEM), Opti−MEM/Reduced−Serum Medium, and phosphate buffer saline (PBS) were purchased from Gibco. FBS (Certified Fetal Bovine Serum) was from biological industries, 04−001−1ACS.

### 2.2. Tube Experiment

The mixture of BSA (30 mg/mL) and D−glucose (0.5 M) was incubated with four different concentrations of L−arginine (0, 36, 180, 360 mM) for 3 months at 37°. Different concentrations of L−arginine (0, 1.56, 6.25, 12.5, 50, 100, 200 mM) were added to the solutions consisting of mice brain proteins (13.9 mg/mL) and MG (20 mM) for 14 days at 37 °C. All reaction systems were added with 1% penicillin–streptomycin solution and sodium azide.

### 2.3. Cell Culture and Treatment

N2a cells were cultured in high glucose DMEM: Opti−MEM (1:1) supplemented with 10% FBS at 37 °C in a humidified atmosphere containing 5% CO_2_. To detect the protective effect of arginine, cells were pre−treated with arginine (6.4 mmol) or PBS for 2 h, and then cells were cultured with MG (1 mmol) or vehicle solutions (Control group), (MG group), (MG+ Arg−group) for 2 h, and cells were harvested for further examination.

### 2.4. Animals and Treatment

The 3 × Tg mice (PS1m146v/APPswe/TauP301L) were bought from the Jackson Laboratory, and C57 mice were purchased from the Experimental Animal Center of Tongji Medical College. All mice were housed in cages (3−5 mice per cage) and were kept under standard conditions: 12 h light–dark cycles, 22 °C, and libitum access to food and water. All animal practices were conducted strictly by protocols approved by the Institutional Animal Care and Use Committee, and the Academic Review Board of Tongji Medical College, Huazhong University of Science and Technology approved the animal study. The 3 × Tg mice, classic AD model mice, presented with progressive neuropathology, plaques, LTP/LTD changes, tangles, and exacerbating cognitive impairments. Of note, they exhibited memory deficits and learning damage as early as the 4th month, which preceded the formation of plaques or tangles. Therefore, it was a good model to explore the factors triggered and/or aggravated pathological changes. The 3 × Tg mice, at 6 months old, were randomly divided into three groups: Arg−high group (0.5 mg/g of body weight), Arg−low group (0.2 mg/g of body weight), and Saline group receiving L−arginine or saline intra−peritoneally (I.P.) every second day for 45 days. Both dose injections of L−arginine were reported within the safe range [34,35,36]. All mice performed the Novel Object Location test and Morris Water Maze (MWM). The next day, mice were euthanized with isoflurane and brains were quickly removed.

### 2.5. Morris Water Maze

The Morris Water Maze (MWM) is composed of two parts, a training phase and probe test phase. During the training, mice (WT, *n* = 5; 3 × Tg, *n* = 5; Arg−L, *n* = 8; Arg−H, *n* = 8) were trained to find the submerged platform according to the cues outside the maze for 60 s in a round water tank in which the water was mixed with milk powder to hide the platform. Once the mice found and stayed on the platform for no less than 3 s, the trial ended. If subjects failed to reach the platform within limited time, the practitioner guided them to the platform and kept them there for 20 s. Each mouse underwent three trials per day for 5 consecutive days. The escape latency (time to find the hidden platform) and swimming path of each subject were recorded by a digital device. During the probe trial, the platform was removed; mice swam for 60 s, and data such as time point to the first crossing platform, swimming speed, time in the target quadrant, and platform crossing numbers were recorded by computer. Practitioners were blinded and data were analyzed by another person.

### 2.6. Novel Object Location

The Novel Object Location protocol was from another study, with minor adjustment [37]. It consisted of three sessions: habituation, training, and test sessions. On the first day, mice (WT, *n* = 10; 3 × Tg, *n* = 8; Arg−L, *n* = 10; Arg−H, *n* = 9) were habituated to a quadrate white opaque chamber which contained no objects but spatial cues for 5 min. The chamber was cleaned by 75% ethanol before habituating next subject. Forty−eight hours after the habituating, it was the training session. The chamber was populated with object A and object B, and then mice entered the chamber sequentially to get familiar with both objects for 5 min. Again, 75% ethanol was used for cleaning. On the next day, in the test session, the location of object A was displaced to another location and mice were given 5 min for free exploration. Time exploring objects A and B in the last two days was recorded by video tracking system. Subjects exploring less than 15 s were excluded. Practitioners were blinded and data were analyzed by another person; for detailed analytic methods, refer to article [37].

### 2.7. Western Blot and Dot Blot

N2a cells were gathered and lysed in RIPA Lysis buffer (weak) (P0013D, Beyotime, China), homogenizing at 4 °C. Dissected hippocampi were homogenized in RIPA Lysis buffer (strong) (P0013B, Beyotime, Shanghai, China) at 4 °C. These homogenates were boiled for 10 min before ultrasonic waving treatment (60 Hz) and then centrifuged (12,000× *g* rpm) for 10 min. Next, the protein level (BCA kit) of the supernatants was collected and analyzed. Samples adding 10% β−mercaptoethanol and 0.05% bromophenol blue were boiled again for 10 min. Proteins in samples were separated by 10% or 8% SDS polyacrylamide gel electrophoresis (SDS−PAGE) and subsequently transferred to nitrocellulose membranes. The method of dot blot we conducted was according to the article from Prof. Jin−Jing Pei [38]. First, we drew a grid on nitrocellulose membrane to mark the area to be blotted. A 3 μL spot of samples, which adapted to the same protein concentration, was dropped in the center of the grid by a narrow−mouth pipette tip. Then, the membrane was kept at 37 °C to dry. Nitrocellulose membranes were blocked by 5% skim milk in PBS for 1 h at room temperature, followed by incubating with primary antibody at 4 °C overnight. A second antibody (anti−rabbit or anti−mouse) was incubated with the membrane for 1 h at room temperature. The blots were visualized via the enhanced chemiluminescent (ECL) imaging system (610007−8Q, Clinx Science Intrument CO., Ltd., Shanghai, China) or Odyssey Infrared Imaging System (LI−COR Bioscience, Lincoln, Nebraska, USA). WB and DB are calculated with software, image J and GraphPad Prism 8.

### 2.8. Immunoprecipitation

Cell lysis buffer for western or IP (P0013, Beyotime, Shanghai, China) was used to homogenize tissues or cells at 4 °C for 30 min. Lysates were centrifuged at 12 000× *g* for 10 min and then the supernatants were collected. Roughly 400 μg of total proteins in the volume of 200 μL were rotated with 4 μg anti−bodies at 4 °C overnight. The following day, 40 μL protein A + G agarose (P2055, Beyotime, Shanghai, China) was added and rotated at 4 °C for 6 h. The agarose beads were washed three times, re−suspended, and boiled in 40 μL SDS−loading buffer in sequence. The supernatants were collected for further examination.

### 2.9. Statistical Analysis

All data are expressed as means ± SD. The significance of data in two conditions was evaluated by the two−tail Student’s unpaired *t* test, and comparisons of multiple groups were assessed by the one−way ANOVA of variance or two−way analysis of variance procedure followed by post hoc Tukey’s test. Significance point was set at *p* < 0.05. All experiments were conducted at least three times.

## 3. Results

### 3.1. The Level of Glycation in γ_2_ Subunit of AMPK Increased and the Level of pT172−AMPK Decreased in 3 × Tg Mice

Insulin resistance and glucose metabolism disorders are broadly presented among AD brains [39]. To further confirm whether AMPK participates in the pathology of AD brains, AMPK (AMPK α) and the activated marker pT172 (pThr172−AMPK) were detected by western blot. The data showed that the level of pT172 is down−regulated in triple transgenic (3 × Tg) mice brains compared with wild−type (WT) ones, which indicated that AMPK activity is destroyed in 3 × Tg mice (Figure 1A,B). Taking aberrant AMPK activity into consideration, we first examined glycation levels in 3 × Tg mice brains by dot blot, according to the theory that glycation might disorganize the structures of proteins and disturb their physical functions [20], and found elevated glycation levels compared to WT mice (Figure 1C,D). Given the predominant allosteric regulatory role of the γ−subunit in AMPK, we speculated that the γ subunit of AMPK was highly glycated and structurally altered in AD. To validate this, we conducted immunoprecipitation of AMPK γ_2_ subunit (main gamma isoform expressed in the brain), followed with western blot with antibody. We found that the level of glycation in the γ_2_ subunit of AMPK was significantly enhanced in AD model mice (Figure 1E) in contrast to WT mice, suggesting that the γ_2_ subunit is significantly glycated, and at the same time, pT172−AMPK declines in 3 × Tg mice.

### 3.2. Arginine Decreases the Glycation of Proteins in a Dose−Dependent Manner

MG is one of the major carbonyl species resulting in the formation of AGEs [40], fitting with the claim that arginine−directed glycating agents account for the majority of AGEs in both cellular and extracellular proteins [19], suggesting that MG has a preference for arginine sites. Moreover, some evidence has shown a protective role of lysine on lysozyme glycation in T2DM patients [41]. We hypothesized that arginine competitively protected proteins from glycation by carbonyl groups of reducing sugars, especially MG. To test the hypothesis, we performed a tube experiment. Arginine at various concentrations was added to the tube that contains the mixture of BAS and D−glucose. The image of dot blot showed that reduction of AGEs occurred in an arginine−dosage−dependent manner (Figure 2A,B). To further assess the effect of arginine on protein glycated by MG, the mixture system of MG and mice brain proteins co−incubated with arginine over a range of concentrations. We observed that the extent of glycated proteins declined as the concentration of arginine increased (Figure 2C,D). In addition, we found that reduction of AGEs level occurred in an arginine−concentration−dependent manner in MG−induced AGEs reaction system (Figure 2E,F). Together, arginine binds to substances like MG and decreases glycation of proteins over a range of concentrations.

### 3.3. Arginine Mitigates Glycation in γ_2_ Subunit of AMPK Induced by MG and Maintains the AMPK Function in N2a Cells

To assess the biological effect of arginine on glycation, we cultivated N2a cells in media with arginine (6.4 mmol) or PBS for 2 h and exposed cells to MG (1 mmol) for 2 h, and harvested N2a cells for further examinations. Similar effects of arginine on glycations were observed in N2a cells. We noticed that intracellular proteins were less glycated in the Arg + MG treatment group compared to the MG group (Figure 3A,B). Also, immunoprecipitation showed that arginine reduced the glycation in the γ_2_ subunit of AMPK (Figure 3C). In alignment with these findings, compared to the MG group, Arg + MG restored the activity of AMPK according to the increased level of pT172−AMPK by western blot (Figure 3D,E) in N2a cells. In order to further detest the function of AMPK, we examined the level of pSer79−ACC (acetyl—CoA carboxylase) since it is a well−known downstream of AMPK [42,43,44]. As we expected, arginine rescues AMPK function indicated by a higher level of pSer79−ACC compared Arg + MG group to MG group (Figure 3D,F). Taken together, arginine decreases the glycation of γ_2_ subunit of AMPK, meanwhile, rescues AMPK function in MG−treated N2a cells.

### 3.4. Arginine Defends γ_2_ Subunit of AMPK against Its Glycation and Improves AMPK Function in 3 × Tg Mice

We want to know whether decreasing glycation of AMPK will rescue its activity and henceforth improve pathologies in AD. The image of dot blot displayed that arginine reduces glycation of hippocampal proteins dose−dependently comparing Arg groups with Saline group in 3 × Tg mice (Figure 4A,B). Next, we examined the effect of arginine on glycation in γ_2_ subunit of AMPK, and the result of immunoprecipitation indicated that the glycation level in γ_2_ subunit of AMPK was significantly down−regulated in Arg group (Figure 4C). Consistent with these findings, western blot demonstrated that administration of arginine reversed deficits of AMPK activation (Figure 4D,E) and improved AMPK function, as suggested by the up−regulated phosphorylation of the Ser79 site in ACC (Figure 4D–F). Profoundly, we conducted western blots on several typical pathological markers of AD, and compared with the Saline group, the Arg treatment group greatly down−regulated the expression of p−Tau on Ser 404 (Figure 4G–I), even though there was only a slight decrease in p−Tau on Ser 396 (Figure 4I). Together, arginine protected AMPK γ_2_ from being glycated, preserved the activation of AMPK, and ameliorated AD pathologies in 3 × Tg mice.

### 3.5. Arginine Attenuates the Impairments of Hippocampal−Dependent Spatial Learning in Memory in 3 × Tg Mice

The hippocampus is one of the key elements in spatial learning and memory [45]. Comprehensive studies have shown hippocampal−dependent spatial learning and memory are deeply destroyed in 3 × Tg mice. Morris Water Maze (MWM) and the object location task are behavior tests to evaluate cognitive abilities [46]. The MWM data indicated that the escape latency time of Arg groups was significantly shorter than Saline group at day 5 during the 5 days of consecutive training, which illustrated arginine treatment alleviated learning ability damage of 3 × Tg mice (Figure 5B). The probe trial elicited higher platform crossings in the Arg groups compared to the Saline group (Figure 5C) as well as shorter escape latency (Figure 5D). Swimming speeds among all the groups were comparable, which indicated that these differences are independent of speed factor. (Figure 5E). Similarly, we conducted an object location task on these objects. Aligned with the MWM test, the Arg group showed remarkable amelioration in cognitive impairments, shown by the higher exploring time with the object in a new location (Figure 5G). These data support administration of arginine as effective for rescuing cognitive deficits in 3 × Tg mice.

## 4. Discussion

We found that the level of p−AMPK was down−regulated in 3 × Tg mice compared to WT mice, which is consistent with other AD−like pathological mice models, such as APP/PS1 and STZ−induced AD−like mice [47,48]. The imbalance of AMPK function aggravates oxidative stress [49], inflammation [50], insulin resistance, and accumulation of Aβ and p−Tau [51,52], all of which will accelerate pathological courses of patients with AD. However, the mechanism of AMPK dysfunction is unknown in AD. Here, we found that the hippocampal glycation level was elevated in 3 × Tg mice in contrast with WT mice. Hence, we assumed that overwhelmed glycation in AMPK might disturb its function.

Abnormal accumulations of AGEs have been verified in human AD and DM brains [53,54]. Studies show that AGEs contribute vastly to the progression of AD and pathologies of diabetes mellitus (DM) and its complications. Glycation takes place at three positively charged amino acid residues: arginine, lysine, and histidine. Glycated proteins exhibit alterations in physical functions or even their biological ones, which might result from charge neutralization [55], hidden specific amino acid binding sites, and so on. These variants destroy protein functions. Interestingly, AMPK, an RD (Arg–Asp) kinase, stabilizes the AL through charge interactions [15], which means that proper charges in AMPK subunits are significant to its function. Besides, AGEs are closely related to insulin resistance according to human research [56]. Further, glycated proteins could trigger fierce radical oxidative stress and inflammation [57,58]. Therefore, glycated proteins may result in insulin resistance.

It has been shown that AMPK dysfunction accelerates pathologies of DM and its complications [59]. Further, targeting the activation of AMPK mitigates DM and its complications [60]. Consequently, AMPK dysfunction is the converging point of AD and DM. AMPK conformational changes regulate phosphorylation of its kinase activation loop (AL), either activating or inhibiting its function. The γ subunit of AMPK directly binds to AMP/ADP and exaggerates AMPK AL up to 10−fold [14]. Of note, it has been reported that MG obstructed AMPK function by binding to γ subunit [61]. Hence, we proposed that the glycated γ_2_ subunit of AMPK destructed its activity, which aligned with our results showing higher glycation in the γ_2_ subunit of AMPK and lower activity in AMPK.

γ subunit isoforms of AMPK among species as well as tissues share four conserved repeated sequences, termed cystathione β−synthetase (CBS1−4) motifs, of which CBS 1 and 3 are key mediators of allosteric regulation of AMPK [62]. Mounting reports have indicated that several arginine sites of CBSs are of great importance and that mutations of any of these will cause AMPK dysfunction [63,64]. Therefore, we postulated glycation of γ subunit, especially on the critical arginine residues, disturbing peptide charges damages its function. Given that the case–cohort study presented lower levels of arginine among T2DM patients, it also underlined the importance of arginine bioavailability [65], and the study showed that L−lysine reduced lysozyme glycation in T2DM patients [41]. We presumed that arginine could prevent the arginine motif of the γ subunit from glycation and attenuated AD−like pathology. Our data verified that arginine impedes protein glycations in vitro and in vivo and that administration of arginine decreases glycation in the γ_2_ subunit of AMPK and improves AMPK activity. On the other hand, arginine is involved in the regulation of brain function, such as ameliorating AD brain pathologies in multiple ways. For instance, arginine is a precursor of nitric oxide, which exerts neuroprotective effects by decreasing vascular stress, protecting endothelial cells, and increasing brain perfusion [66], which protect blood–brain barrier integrity and provide adequate nutrients. Arginine plays an important role in immune responses in periphery. Also, in the central nervous system, arginine has a close relationship with immune reactions. Increased arginine bioavailability remarkably improves pathology and cognitive function by promoting local immunity of microglia [67]. Furthermore, arginine participates in nitric−oxide−mediated processes such as neurotransmission [68] and dopamine transport [69] and modulates the excitability of neuronal states [70]. We could not exclude any other effects of arginine on 3 × Tg mice, but our results indicate that arginine decreases the glycation of the γ_2_ subunit in AMPK and mitigates cognitive impairment in 3 × Tg mice.

The proliferative cell line N2a is a mouse neuroblastoma cell line, frequently used in neuroscience research. N2a cells could differentiate into neurons under certain conditions. Differentiated and undifferentiated N2a cells have some variances in neuronal differentiation, axonal growth, and neurotransmitters. Here, we explored the glycation of AMPK and the glycation effects on AMPK function in undifferentiated N2a cells. This model had limitations. We could not observe neural morphologies, axonal growth, etc. However, the Millard reaction is a non−enzymatic reaction, which means it could react once substrates are available. Further, much data have shown that the glycation of proteins is a universal phenomenon in different tissues and cells [71,72,73]. We think that undifferentiated N2a cells react highly similarly with neurons in protein glycation when cells suffer from metabolic disorder. Moreover, our data display the same change between N2a cells and mice.

Further studies addressing specific glycation sites leading to AMPK dysfunction are required for direct evidence that non−glycated AMPK reverses functional deficits as well as insulin resistance. Although many mechanisms need further exploration, our results completely support lower glycation in the γ_2_ subunit of AMPK as exhibiting higher AMPK activity.

## 5. Conclusions

Our study showed that arginine mitigates AMPK glycation, restores AMPK activity, and improves cognitive impairments in 3 × Tg mice, suggesting that glycation of the γ_2_ subunit is involved in allosteric regulation of AMPK and arginine reverses AMPK malfunction and AD−like pathological and behavioral abnormalities in 3 × Tg mice. These findings might suggest a new road for AD therapy.

## Figures and Tables

**Figure 1 cells-11-03520-f001:**
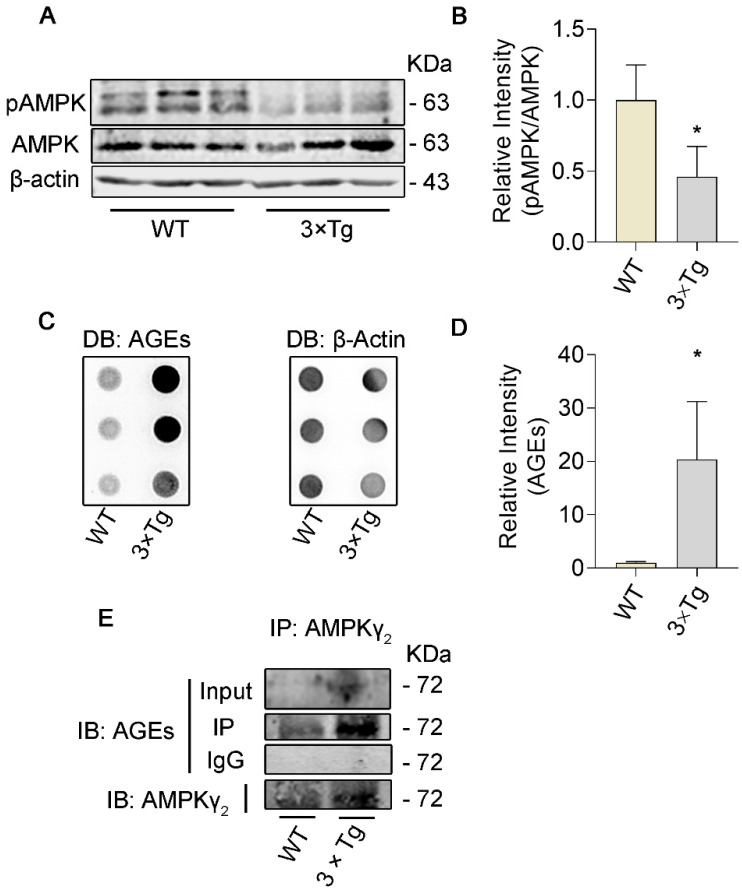
The level of glycation in γ_2_ subunit of AMPK is increased and the level of pT172−AMPK is decreased in 3 × Tg mice. The hippocampal proteins are extracted from 7.5−month−old 3 × Tg mice and wild−type (WT) mice. (**A**,**B**) The expression and quantitative analysis of pT172−AMPK and AMPK were examined by western blot in the hippocampi of 3 × Tg and WT mice. (**C**,**D**) The level of advanced glycation end−products (AGEs) of brain proteins in 3 × Tg and WT mice were detected by dot blot. (**E**) Hippocampal extracts were immune−precipitated by antibody against AMPK γ_2_ and then detected with antibody against AGEs. All data are presented as the means ± SD (*n* = 3), statistical analysis with Student’s unpaired *t* test; * *p* < 0.05 vs. WT mice.

**Figure 2 cells-11-03520-f002:**
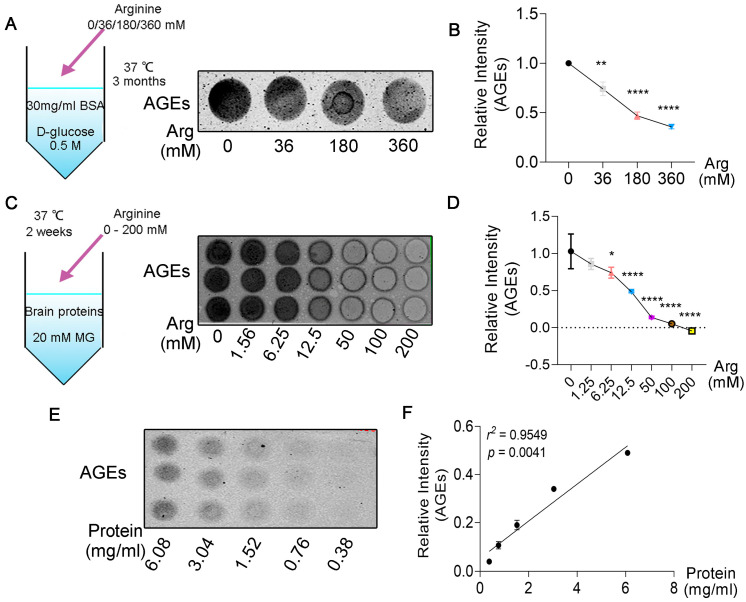
Arginine decreases the glycation of proteins in a dose−dependent manner. (**A**,**B**) Bovine serum albumin (BSA, 30 mg/mL) and D−glucose (0.5 M) were incubated with L−arginine (0, 36, 180, 360 mM) at 37 °C for 3 months. Dot blot was used to measure the level of glycation with antibody against AGEs. (**C**,**D**) Mice brain proteins (13.5 mg/mL), methylglyoxal (MG, 250 mM) were reacted with L−arginine (0, 1.56, 6.25, 12.5, 50, 100, 200 mM) at 37 °C for a fortnight. Dot blot was performed. (**E**,**F**) Sequentially diluted samples from (**C**) (proteins + MG + Arg (6.25 mM)) were detected through dot blot with antibody against AGEs and assessed by correlation analyses. Significant correlation (*r*^2^ = 0.9549, *p* = 0.0041). Data are exhibited as mean ± SD (*n* = 3). (**B**,**D**) statistical analysis with one−way ANOVA with a post hoc Turkey’s test. * *p* < 0.005, ** *p* < 0.001, **** *p* < 0.0001 vs. 0−Arg group.

**Figure 3 cells-11-03520-f003:**
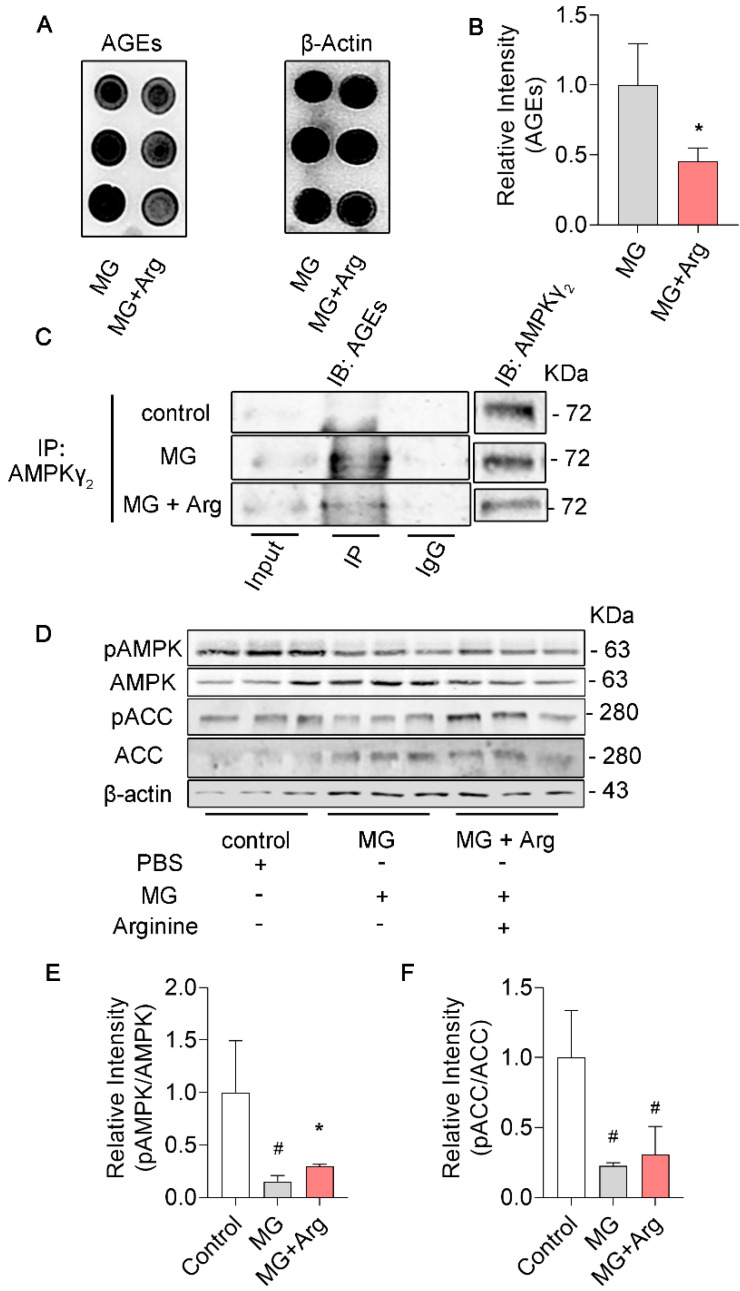
Arginine mitigates glycation in γ_2_ subunit of AMPK induced by MG and maintains the AMPK function in N2a cells. (**A**,**B**) The glycated proteins were detected and analyzed by dot blot with antibody against AGEs. (**C**) The N2a lysates were immune−precipitated with antibody against AMPK γ_2_, followed by western blot with antibody against AGEs. (**D**–**F**) The level of pT172−AMPK, AMPK, pS79−ACC, and ACC were measured and analyzed by western blot in MG, MG + Arg groups. All data are presented as the means ± SD (*n* = 3), statistical analysis with one−way ANOVA with a post hoc Turkey’s test. * *p* < 0.05 vs. MG; # *p* < 0.05 vs. control.

**Figure 4 cells-11-03520-f004:**
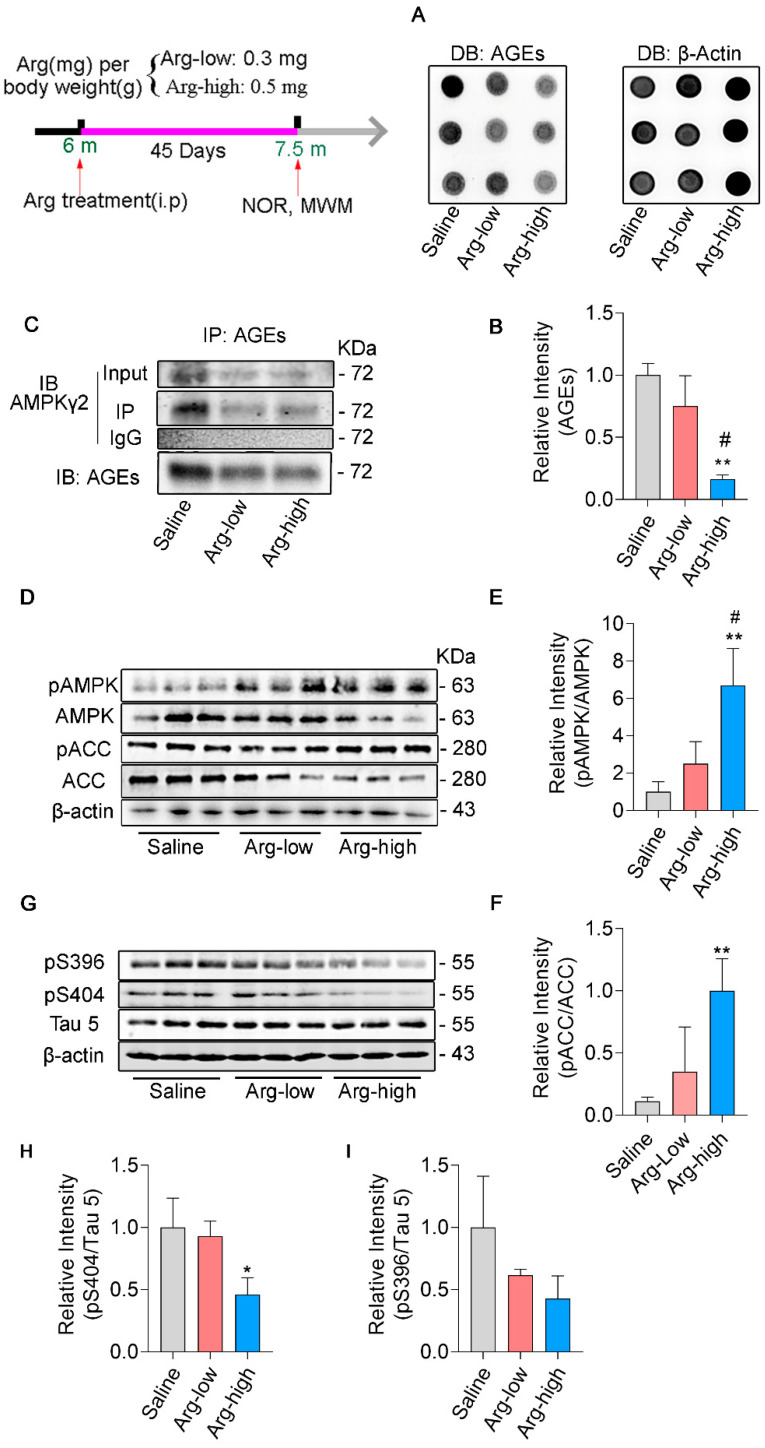
Arginine defends γ_2_ subunit of AMPK against its glycation and improves AMPK function in 3 × Tg mice. The hippocampi were extracted from 3 × Tg Saline group (Saline), 3 × Tg Arg−low group (Arg−low), 3 × Tg Arg−high group (Arg−high) at 7.5−month−old. (**A**,**B**) The levels of hippocampal AGEs were examined by dot blot with antibody against anti−AGEs. (**C**) Dissected hippocampi were immune−precipitated with antibody against AGEs and then examined by western blot with antibody against AMPK γ_2_. (**D**–**I**) The expressions of AMPK, pT172−AMPK, pS79−ACC, ACC, Tau 5, pS396, and pS404 were examined by western blot. All data were exhibited as the means ± SD (*n* = 3), statistical analysis with one−way ANOVA with a post hoc Turkey’s test. * *p* < 0.05, ** *p* < 0.01 vs. Saline group, # *p* < 0.05 vs. Arg−low group.

**Figure 5 cells-11-03520-f005:**
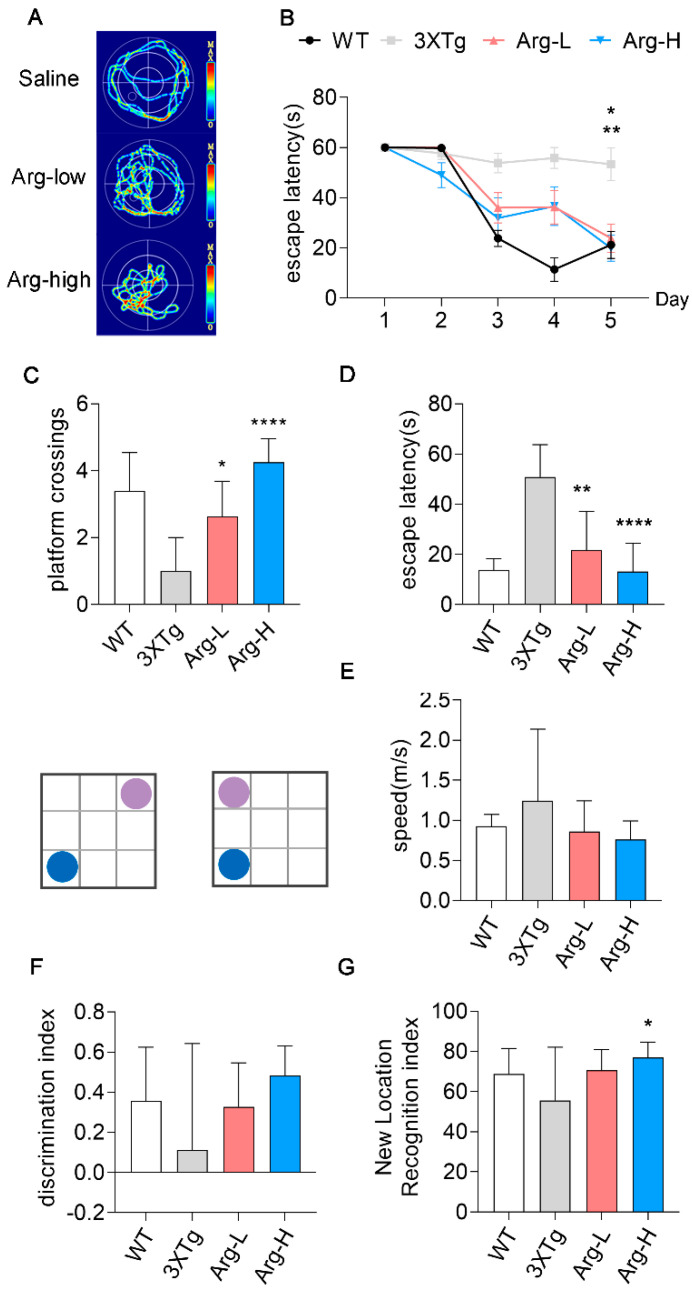
Arginine attenuates the impairments of hippocampal−dependent spatial learning in memory in 3 × Tg mice. In the MWM test, mice (WT, *n* = 5; Saline group, *n* = 5; Arg−low, *n* = 8; Arg−high, *n* = 8) underwent 5 consecutive days for learning ability tests. (**B**) The latencies to reach the hidden platform were recorded each day. In the probe trial, the latency time of first reaching the platform (**D**), numbers of platform crossings (**C**), swimming speed (**E**), and representative swimming routes (**A**) were recorded. (**F**) The discrimination indexes were recorded. (**G**) Arg−high treatment increased the novel location recognition index in 3 × Tg mice. Data were expressed as mean ± SD (*n* = 3). (**B**) was analyzed by two−way repeated−measures ANOVA with a post hoc Bonferroni’s test. (**C**,**D**) were measured by one−way ANOVA with a post hoc Tukey’s test. * *p* < 0.05, ** *p* < 0.01, **** *p* < 0.0001 vs. 3 × Tg.

## Data Availability

The raw data supporting the conclusion of this article will be made available by the authors, without undue reservation.

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
