# Peer review of "Arginine Reduces Glycation in γ2 Subunit of AMPK and Pathologies in Alzheimer’s Disease Model Mice"

_cells, 2022, doi:10.3390/cells11213520_

Round 1
Reviewer 1 Report
In this work, the authors provide evidence that AMPK gamma2 subunit undergoes glycation, and this event significantly impacts AMPK activity. Importantly, gamma2 subunit is highy glycated in a preclinical model of AD, and arginine administration is able to prevent glycation, alleviating behavioral symptomatology. The manuscript shows interesting data but presents several concerns that need to be addressed:
1) Data are sometimes expressed as the mean ± S.E.M. The standard error of the mean indicates the uncertainty of how the sample mean represents the population mean. In my opinion, the authors inappropriately report the SEM instead of the Standard Deviation (SD). Since the SEM is always less than the SD, it deceives the reader into underestimating the variability between individuals within the study sample.
2) The number of biological replicates should be reported in each figure legend
3) A housekeeping protein should be always shown for Western blot analysis. In some cases, it is not visualized (figure 1A, 3D, 4D). Since the expression of proteins of interest (i.e. AMPK, ACC) could change in dependence on the pysiopathological context, it is necessary to show a housekeeping protein to better sustain the equal loading of the samples.
4) Figure 1A: p-AMPK generally appears as a single band. In this representative immunoblot, it is shown as a double band. The authors should explain this inconsistency.
5) N2a cells can be considered a model of neuronal cell only when they are differentiated. Considering the information included in the manuscript, the authors used undifferentiated N2a. Thus, the main experiments on N2a should be repeated on differentiated, neuronal-like cells.
6) The authors ascribe the positive effects of arginine treatment to the rescue of AMPK activity. For instance, they stated that "The MWM data indicated, the escape latency time of Arg-groups started significantly shorter than Saline-group at the day 5 during the 5 days consecutive trainings, which illustrated arginine treatment improved AMPK activity and alleviated learning ability damage of 3 × Tg mice.". Additionally, they conclude that "We noted that arginine presents glycation protective advantages toward AMPK in 3× Tg mice model". The whole manuscript is disseminated by these statements, which should be modified as they are too speculative. Notably, the involvement of AMPK in arginine effects is only correlative, as mechanistic experiments aimed at demonstrating the causality of the events are completely lacking. In other words, the beneficial effects exerted by arginine could be mediated, at least in part, by other pathways different from AMPK. The authors should revise the entire work and be more restraint in their interpretations.
7) English language should be carefully revised as several grammar and syntax errors are present, which sometimes severely undermine the readability of the manuscript.
Author Response
We would like to thank you for thoroughly reviewing our manuscript and making many thoughtful comments. We were very pleased to see you recognized the novelty and potential significance of our work. Here are our point-by-point responses:
Point 1
Response 1:
We agree with reviewer’s opinion about SEM and SD. When data exist in great differences in samples, we usually use SEM to replace SD for presenting variability. (Heckmann BL, etc. Cell. 2019), (Pan RY, etc. Cell Metab. 2022) Of course, we also do not understand it completely, just follow the presenting format. So we have made some corrections.
Point 2
Response 2:
Yes, the number of samples should be described. So we have added the number of biological replicates in each figure legend.
Point 3
Response 3:
Yes, it is a good suggestion. We have added the housekeeping proteins to the above figures.
Point 4
Response 4: In figure 1A, pAMPK shows two distinct bands, and in figure 3D and 4D, pAMPK also has two bands if we look closely, but one of the bands is particularly shallow. There is a similar phenomenon about image of WB in other paper. (Kun-Liang Guan, et. al. Cell. 2006). We think that this inconsistency might be some factors, such as the fluctuations in the amount of sample loads, the efficacy of antibodies and exposure of NC. We repeated and changed the assay condition, but we have not obtained same image of WB.
Point 5
Response 5: It is a good question. Undifferentiated N2a and differentiated N2a have some variances in neuronal differentiation, axonal growth, and so on. Here, we aim to explore the glycation of AMPK and effects of AMPK glycation on its function and cell change, a lot of evidence has shown that glycation of protein is a universal phenomenon in different tissues and cells. So we think that undifferentiated N2a cells react highly similar to neurons in protein glycation when cells suffer from metabolic disorder. Of course, differentiated N2a cells may be more similar to neurons, but our data also display the same change between N2a cells and mice.
Point 6
Response 6: Yes, we agree with reviewer’s opinion. To avoid over-speculation, we have revised the related words and marked in red words.
Point 7
Response 7: We appreciated your advice and revised the manuscript in grammar and syntax.
Reviewer 2 Report
Zhun et al revealed arginine reduces glycation in γ2 subunit of AMPK and ame-2 liorates pathologies of Alzheimer's Disease via model mice, suggesting glycation of γ2 subunit as a new target for AD therapy. It is an interesting manuscript. But there are some limitations that need to be well addressed:
1. Title makes the reader a little confused, please improve it
2. Introduction needs to make a clearer connect AD vs T2DM, in paragraph 2, it makes unclear meaning, please revise accordingly
3. Why authors chose to test the two behavior studies? I think an additional as Barnes maze test needs to be involved
4. Some Ig biomarkers for the mice model should be additionally tested and imaged as confocal. Also, a Western blot from mice tissues is needed as well. Both additional data need to come from control and treatment groups.
5. A better discussion is needed to show some potential pathogenic and therapeutic pathways as the authors stated
6. English writing style still needs to be improved more since many sentences and errors writing repeat
Author Response
Thank you for your precious comments and advice. Those comments are all valuable and very helpful for revising and improving our paper. We have studied comments carefully and have made correction which we hope meet with approval. The responds to the comments are as flowing:
Point 1
Response 1:
Thank you for your opinion. We have cut out some lengthy words in the title to make us better understood.
Point 2
Response 2:
It is good advice. We state the relationship between AD and T2DM, but a few detailed information to support the statement, so we added more evidences in paragraph 2.
Point 3
Response 3:
We chose these two behavior studies because novel location recognition (NLR) examines cognition, especially spatial memory and discrimination and morris water maze (MWM) is to test spatial and long-term memory. Compared to Barnes maze (BM), which also aims at examing spatial and long-term memory, MWM would provide us more solid results since our lab has a mature experimental procedure and equipment for MWM. Therefore, we conducted MWM and BM for cognition tests.
Point 4
Response 4:
I am so sorry that I do not understand what the “Ig biomarker” refers to, so I cannot make a further explanation and improvement. I would appreciate it if you could give me a more detailed description.
Point 5
Response 5:
It is a valuable suggestion. We have elaborated the potential pathogenic pathways in AD model and arginine’s therapeutic pathways in the discussion.
Point 6
Response 6:
We appreciated your advice. The manuscript has been revised and corrected carefully.

Round 2
Reviewer 2 Report
Thanks for addressing almost my concerns. However, there is a comment that needs to be fully addressed before accepting. "4. Regards AMPK dysfunction and its associated Tau hyper-phosphorylation and Aβ deposits, I suggested additional experiments to conclude Tau phosphorylation level in AMPK down/upregulation mice through brain slice immunostaining and quantitative analysis".
Author Response
It is thoughtful thinking. And considering our previous work, we have proved that upregulation of AMPK ameliorated Tau phosphorylation and vice versa (photos are displayed in the following pdf file, Lin Wang, Molecular Neurobiology, 2020). Here, the results in figure 4 from wb experiments could validate our conclusion that activating AMPK attenuates Tau hyperphosphorylation.
